# Precise and Accurate DNA-3′/5-Ends Polishing with *Thermus thermophilus* Phage vb_Tt72 DNA Polymerase

**DOI:** 10.3390/ijms252413544

**Published:** 2024-12-18

**Authors:** Sebastian Dorawa, Tadeusz Kaczorowski

**Affiliations:** Laboratory of Extremophiles Biology, Department of Microbiology, Faculty of Biology, University of Gdansk, 80-308 Gdansk, Poland; sebastian.dorawa@ug.edu.pl

**Keywords:** PolA-type enzyme, proofreading, 3′→5′ exonuclease activity, molecular cloning

## Abstract

Tt72 DNA polymerase is a newly characterized PolA-type thermostable enzyme derived from the *Thermus thermophilus* phage vB_Tt72. The enzyme demonstrates strong 3′→5′ exonucleolytic proofreading activity, even in the presence of 1 mM dNTPs. In this study, we examined how the exonucleolytic activity of Tt72 DNA polymerase affects the fidelity of DNA synthesis. Using a plasmid-based *lacZα* gene complementation assay, we determined that the enzyme’s mutation frequency was 2.06 × 10^−3^, corresponding to an error rate of 1.41 × 10^−5^. For the exonuclease-deficient variant, the mutation frequency increased to 6.23 × 10^−3^, with an associated error rate of 4.29 × 10^−5^. The enzyme retained 3′→5′ exonucleolytic activity at temperatures up to 70 °C but lost it after 10 min of incubation at temperatures above 75 °C. Additionally, we demonstrated that Tt72 DNA polymerase efficiently processes 3′/5′-overhangs and removes a single-nucleotide 3′-dA overhang from PCR products at 55 °C. These characteristics make Tt72 DNA polymerase well suited for specialized molecular cloning applications.

## 1. Introduction

The primary function of DNA polymerases is DNA synthesis, where they extend the primer strand in the 5′→3′ direction by adding complementary deoxyribonucleotides to the 3′OH end. In addition, these enzymes may possess 3′→5′ exonuclease (exo) activity, which corrects errors by removing misincorporated nucleotides, as well as 5′→3′ exo activity, which degrades RNA primers [1]. Beyond these core functions, DNA polymerases can act as 5′-deoxyribose-phosphate lyases [2] or primases [3]. They also play a key role in several DNA repair mechanisms, including mismatch repair (MMR), nucleotide excision repair (NER), base excision repair (BER), and the repair of double strand breaks via non-homologous end joining (NHEJ) or homologous recombination (HR) [4].

The striking functional versatility of DNA polymerases has made them indispensable in molecular biology, driving advancements in both research and technology [5]. These enzymes are integral to molecular diagnostics and have been key to developing numerous specialized life science technologies. Among the most widely used thermophilic DNA polymerases is Taq DNA polymerase from *Thermus aquaticus* [6]. Its exceptional thermal stability has made it a cornerstone of PCR technology, revolutionizing molecular biology by enabling rapid and efficient DNA amplification [7,8]. Another notable enzyme is DNA polymerase I from *Thermus thermophilus*, which exhibits reverse transcriptase activity in the presence of Mn^2+^ ions, making it a versatile tool for various molecular applications [9]. Archaeal DNA polymerases, primarily derived from microbes within the *Pyrococcus* and *Thermococcus* genera, are also extensively utilized for PCR-based DNA amplification due to their exceptional accuracy in DNA synthesis. This high fidelity stems from their intrinsic proofreading ability, facilitated by 3′→5′ exo activity, which removes misincorporated nucleotides to maintain precise DNA replication [10]. Additionally, archeal polymerases exhibit markedly superior thermal stability compared to their bacterial counterparts. A notable example is DNA polymerase I from *Pyrococcus furiosus*, which, with its proofreading activity, achieves replication fidelity approximately ten times higher than that of Taq polymerase [11].

Phage DNA polymerases are essential tools in various molecular techniques, with DNA polymerases derived from bacteriophages T4, T7, and φ29 being among the most prominent. Although T4 and T7 polymerases belong to different subfamilies—PolB and PolA—they share important functional characteristics. Both enzymes possess two functional domains: a nucleotidyltransferase domain and a 3′→5′ exo domain, enabling efficient end-filling at 5′ ends and exonucleolytic removal at 3′ ends of double-stranded DNA fragments [10]. These features also make them valuable in site-specific mutagenesis [11,12]. T4 DNA polymerase is particularly versatile, capable of removing 3′ single-nucleotide (dA) overhangs from PCR products [13], synthesizing and labeling DNA probes through strand displacement [14], and facilitating ligation-independent cloning [15]. Conversely, T7 DNA polymerase is highly effective in synthesizing complementary cDNA strands [16], removing residual genomic DNA during circular DNA purification, and detecting in situ DNA fragmentation caused by apoptosis [17]. A genetically modified version of T7 DNA polymerase, Sequenase 2.0, has become instrumental in Sanger DNA sequencing [18]. Among phage-derived enzymes, φ29 DNA polymerase stands out for its extensive biotechnological applications due to its high replication fidelity [19,20] and robust strand-displacement activity [21]. This makes it particularly suitable for isothermal DNA amplification, including rolling circle amplification (RCA) [22]. Notably, φ29 DNA polymerase’s minimal reverse transcriptase activity allows it to use RNA as a template in RCA, broadening its scope [23]. Additionally, φ29 DNA polymerase is employed in multiple displacement amplification [24], a method now commonly used for whole genome amplification [25]. It is also critical for next-generation sequencing technologies such as single-molecule real-time sequencing (SMRT) [26,27] and nanopore sequencing [28,29].

In our laboratory, we recently discovered and characterized a novel PolA-type DNA polymerase from the *Thermus thermophilus* phage vb_Tt72 [30]. This enzyme is notable as one of only two known DNA polymerases from thermophilic phages. The present study aimed to examine the enzyme’s 3′→5′ exo proofreading activity, replication fidelity, and DNA-end polishing capabilities. Our findings demonstrate that Tt72 DNA polymerase exhibits exceptional accuracy and precision in processing DNA fragments at elevated temperatures. These attributes make it a valuable tool for molecular cloning applications.

## 2. Results

### 2.1. The 3′→5′ Exonuclease Activity of the Tt72 DNA Polymerase

Recently, our group demonstrated that Tt72 DNA polymerase possesses strong 3′→5′ exo activity [30]. In this study, we further evaluated the exo activity. First, we assessed this activity following heat treatment of Tt72 DNA polymerase for 10 min at temperatures ranging from 55 °C to 95 °C. The results showed that the enzyme retained over 95% of its initial 3′→5′ exo activity between 55 °C and 70 °C (Figure 1a). However, activity decreased markedly at 75 °C, dropping to 25%, with only residual activity detected at temperatures above 75 °C. Interestingly, the 3′→5′ exo exhibited greater thermal stability than the nucleotidyltransferase activity of Tt72 DNA polymerase. While the 3′→5′ exo activity declined more gradually at higher temperatures, the nucleotidyltransferase activity experienced a rapid loss of function above 60 °C and became undetectable beyond 75 °C [30]. Subsequently, we investigated the effect of deoxynucleoside triphosphates (dNTP) on the 3′→5′ exo activity of Tt72 DNA polymerase. Our results showed that the 3′→5′ exo activity persisted even at a dNTP concentration as high as 1 mM (Figure 1b).

### 2.2. Fidelity of the Tt72 DNA Polymerase

The effect of 3′→5′ exo proofreading activity on the fidelity of Tt72 DNA polymerase was analyzed using a plasmid-based *lacZα* gene complementation assay [31,32]. The results, summarized in Table 1, indicate that the mutation frequency of Tt72 DNA polymerase was 2.06 × 10^−3^, with an error calculated at 1.41 × 10^−5^. In the control experiment, the error rates for Taq and Pfu DNA polymerases were found to be 4.27 × 10^−5^ and 3.91 × 10^−6^, respectively. The data indicate that the error rate of Tt72 DNA polymerase is approximately three times lower than that of Taq DNA polymerase but nearly four times higher than that of Pfu DNA polymerase. Additionally, we evaluated the fidelity of the 3′→5′ exo-deficient (D78A) variant of Tt72 DNA polymerase. We have previously shown that substituting aspartic acid at position 78 (motif Exo II) with alanine abolishes 3′→5′ exo activity [30]. For this exo-deficient variant, the mutation frequency was 6.23 × 10^−3^, and the corresponding error rate was 4.29 × 10^−5^. This error rate is higher than that for wild-type enzyme and comparable to that of Taq DNA polymerase. Sequencing of 20 clones generated using the wild-type Tt72 DNA polymerase revealed the mutation spectra, showing the predominance of base substitutions and deletions. Specifically, G → A transitions accounted for 90% of the mutations, while C → A transversions and deletions made up 6.67% and 3.33% of the mutations, respectively.

### 2.3. DNA Ends Polishing with Tt72 DNA Polymerase

Next, we evaluated the efficiency of Tt72 DNA polymerase in processing overhangs at the 3′ and 5′ ends of the pUC18 plasmid linearized with the PstI or BamHI restriction enzymes (Figure 2). The 3′-overhang generated by PstI was removed through the 3′→5′ exo activity of Tt72 DNA polymerase, while the 5′-overhang produced by BamHI was filled in by incorporating complementary nucleotides. In both cases, this resulted in linear plasmids with blunt ends. To assess the efficiency of DNA end-polishing, an α-complementation test was performed. The blunt-ended linear plasmids were first ligated and then introduced into *E. coli* DH5α cells via transformation. The end-blunting process caused a frameshift mutation in the α fragment of the *lacZ* gene, leading to a non-functional β-galactosidase enzyme. Consequently, colonies with these mutations appeared white on LA plates supplemented with X-Gal. The ratio of white colonies to the total number of colonies served as an indicator of the efficiency of dsDNA end-blunting.

The results are summarized in Table 2 and Table 3. Analysis of 5′-overhang blunting demonstrated that the Tt72 DNA polymerase has an efficiency comparable to that of T4 DNA polymerase (Table 2). Furthermore, its efficiency surpasses that of Klenow Fragment and Pfu DNA polymerase. The fidelity of the 5′-overhang filling reaction with Tt72 DNA polymerase was also assessed by sequencing 15 white colonies. This analysis revealed that the enzyme incorporated complementary nucleotides with 100% accuracy during the filling-in reaction.

For 3′-overhang polishing, the Tt72 DNA polymerase demonstrated an efficiency comparable with that of T4 DNA polymerase (Table 3), and higher than that observed for the Klenow Fragment. The end-blunting reaction products were further analyzed by DNA sequencing of 15 clones, confirming that in each case, the enzyme consistently removed 4 nucleotides from the 3′ end.

### 2.4. End Polishing of PCR Products with Single-Nucleotide 3′-Overhang

The next objective was to evaluate the efficiency of Tt72 polymerase in removing single-nucleotide (dA) 3′ overhangs from PCR products. The experimental workflow is illustrated in Figure 3. The protocol uses a cassette carrying a chloramphenicol resistance gene. This cassette was amplified using Taq DNA polymerase, which, due to its inherent terminal transferase activity, adds an extra nucleotide to the 3′ ends of the PCR product [33]. The PCR-amplified DNA fragment was then purified and analyzed by sequencing, which confirmed the presence of an additional adenine—at the 3′ ends. Subsequently, the purified PCR product was treated with Tt72 DNA polymerase to remove the 3′ overhangs. To assess the efficiency of 3′ end-blunting, an α-complementation test was employed. Following treatment with Tt72 DNA polymerase, the polished PCR product was ligated into the pUC18/SmaI vector and introduced via transformation into *E. coli* DH5α cells. The efficiency of the double-stranded DNA end-blunting was determined by the ratio of white colonies to all colonies (indicative of successful end-blunting) to the total number of colonies grown on LA plates supplemented with X-Gal.

The results showed a low percentage of white colonies across all tested DNA polymerases (Table 4). Tt72 DNA polymerase exhibited efficiency in removing 3′-overhangs comparable to that of the control enzyme, T4 DNA polymerase. To confirm successful cloning, the white colonies were screened for the presence of the chloramphenicol resistance gene within the inserted DNA fragment. For this verification, all transformants were plated onto LA plates supplemented with chloramphenicol as an additional selective agent. Notably, 95% of the white colonies grew on the selective medium, indicating successful cloning of the antibiotic resistance cassette. Furthermore, sequencing analysis confirmed that the cloned cassette did not retain an extra nucleotide at the 3′-ends, validating the precise end-polishing capability of Tt72 DNA polymerase. This result underscored the enzyme’s reliability in preparing PCR products for downstream applications requiring blunt-ended DNA.

## 3. Discussion

Over 100 phages infecting *Thermus* bacteria have been isolated so far [34,35,36,37,38,39,40,41,42]. However, only two DNA polymerases from *Thermus thermophilus* phages have been characterized in detail. The first is the DNA polymerase from the thermophilic phage *Thermus thermophilus* MAT72 vB_Tt72 [30], and the second is the enzyme from phage G20c [43]. Both DNA polymerases belong to the PolA-type enzymes and share a similar structural organization, comprising two well-defined domains responsible for 3′→5′ exo and nucleotidyltransferase activities. However, bioinformatics analyses of their amino acid sequences show that both enzymes lack the 5′→3′ exo domain.

The 3′→5′ exonucleolytic activity serves a proofreading function, removing misincorporated nucleotides. In the case of Tt72 DNA polymerase, the nucleotide excision mechanism appears to be similar to that of other enzymes from the PolA family [44]. The removal of a misincorporated nucleotide requires the presence of two divalent metal ions (Mg^2+^ or Mn^2+^), which facilitate the deprotonation of a water molecule [45]. This leads to the production of hydroxyl anion, which is essential for nucleophilic attack on the phosphorus atom, forming the phosphodiester bond with misincorporated nucleotide [44]. Our studies show that the Tt72 DNA polymerase exhibits strong 3′→5′ exo activity even in the presence of high concentrations of dNTPs (up to 1000 μM). Moreover, the Tt72 DNA polymerase retains more than 95% of its activity after incubation at 55–70 °C. However, at 75 °C, only about 25% of the 3′→5′ exo activity remains, and minimal activity is detected at temperatures above 75 °C. Interestingly, the 3′→5′ exo activity is more thermally stable than the nucleotidyltransferase activity of Tt72 DNA polymerase at elevated temperatures. We previously reported that above 60 °C, the nucleotidyltransferase activity of Tt72 DNA polymerase rapidly declines, with no detectable activity beyond 75 °C [30]. Strong 3′→5′ exo activity has also been observed in the PolA-type DNA polymerase from the *T. thermophilus* phage φYS40, which is homologous to Tt72 DNA polymerase [37].

DNA polymerases with high replication fidelity are crucial for reducing amplification errors in PCR and during the end-filling of 5′-overhangs. Enzymes with 3′→5′ exo activity generally exhibit higher fidelity than non-proofreading DNA polymerases [46]. The impact of 3′→5′ exo activity on the fidelity of Tt72 DNA polymerase was directly demonstrated by comparing the error rates of the wild-type enzyme and an exo-deficient variant. The error rate of the exo-deficient variant was three times higher than that of the wild-type enzyme. The 3′→5′ exo-deficient variant carries a single amino acid substitution, replacing aspartic acid at position 78 with alanine, which abolishes its 3′→5′ exo activity [30]. The D78 residue corresponds to the D424 residue in canonical PolA-type *E. coli* DNA polymerase I, which forms hydrogen bonds with two water molecules involved in coordinating the metal ion (metal B) essential for 3′→5′ exo activity [44]. *E. coli* DNA polymerase I deficient in 3′→5′ exo activity exhibits reduced replication fidelity [47]. The error rate for exo-deficient Tt72 DNA polymerase is comparable to that of Taq DNA polymerase, whose poor fidelity is attributed to the inactivity of its 3′→5′ exo domain [48]. The error spectrum of Tt72 DNA polymerase in the plasmid-based *lacZ*α gene complementation assay is similar to that of other PolA-type polymerases [49], characterized predominantly by G to A transitions.

Klenow Fragment, or T4 DNA polymerase, is commonly used for filling in 5′-ends and removing 3′-overhangs produced by restriction enzymes [10]. It was also shown that Pfu DNA polymerase effectively polishes 3′ overhangs of dsDNA [50]. Our results demonstrate that the Tt72 DNA polymerase can serve the same purpose, exhibiting high activity at elevated temperatures and in the presence of high concentrations of dNTPs. Many widely used DNA polymerases, such as T7, Taq, Vent, Klenow Fragment, or DNA polymerase I from *E. coli*, are known to display terminal transferase activity [33,51], which results in the addition of extra nucleotides to the 3′ end of a DNA strand, complicating molecular cloning. However, some DNA polymerases, including Tt72, Pfu, and T4, do not exhibit terminal transferase activity and are effective at removing single-nucleotide overhangs introduced during PCR-based DNA amplification [52]. Our results indicate that Tt72 DNA polymerase performs similarly to T4 DNA polymerase in removing single-nucleotide extensions. This property makes this enzyme a valuable alternative for applications where precise end processing is essential, as it helps mitigate the issues caused by DNA polymerases exhibiting terminal transferase activity.

## 4. Materials and Methods

### 4.1. Bacterial Strains, Plasmid, and Materials

*E. coli* strains BL21(DE3)[pRARE], and DH5α were used for protein overproduction and α-complementation assay, respectively. Bacteria were cultivated at 37 °C in Luria broth (LB) or Luria agar (LA) solid medium [10]. LB or LA media were supplemented with ampicillin (Ap, 100 µg/mL) and chloramphenicol (Cm, 34 µg/mL) when necessary. Plasmid pUC18 (Ap^R^) was used in 3′/5′-end polishing experiments. The vector pHSG576 (Cm^R^), used as a source of the *Cam^R^* cassette, was kindly provided by Dr. Manel Camps (University of California, Santa Cruz, CA, USA). Plasmid pSJ3, used for DNA polymerase fidelity assays, was obtained from Dr. Luis Gabriel Brieba de Castro (Center for Research and Advanced Studies of the National Polytechnic Institute, Mexico City, Mexico). Plasmid DNA was introduced into *E. coli* cells following the method of Inoue et al. (1990) [53]. Deoxynucleotide triphosphates (dNTPs), restriction endonucleases PstI, BamHI, and SmaI, T4 DNA ligase, Klenow Fragment, and T4 DNA polymerase were supplied by Thermo Fisher Scientific (Waltham, MA, USA). Pfu DNA polymerase was purchased from EURx (Gdansk, Poland), while RUN DNA polymerase (Taq enzyme) and X-Gal were obtained from A&A Biotechnology (Gdansk, Poland).

### 4.2. Protein Purification

The plasmid pET15b_polTt72 (Ap^R^) was used for the overproduction of Tt72 DNA polymerase [30]. To enhance recombinant protein overproduction, plasmid pRARE (Cm^R^) was used, which carries genes encoding rare host codon tRNAs (Arg, Gly, Ile, and Pro), thereby optimizing codon usage for heterologous expression [54]. *E. coli* BL21(DE3) [pRARE] cells carrying the expression plasmid were cultivated at 30 °C in 1000 mL LB medium until an A_600_ of 0.4–0.5 was reached. Overproduction of Tt72 DNA polymerase was induced by the addition of isopropyl-β-_D_-thiogalactopyranoside (IPTG) to a final concentration of 1 mM. Following 16 h of incubation at 18 °C, the cells were harvested by centrifugation (4 °C; 10,000× *g*; 20 min) and resuspended in 6 mL of buffer A (50 mM NaH_2_PO_4_ (pH 8.0), 500 mM NaCl, 0.1% [*v*/*v*] Triton X-100, 10% [*v*/*v*] glycerol) supplemented with 10 mM imidazole. Cell disruption was performed by sonication (4 °C, 30 bursts of 10 s at an amplitude of 12 µm; MISONIX sonicator XL2020; Misonix Inc., Farmingdale, NY, USA), followed by centrifugation (4 °C; 10,000× *g*; 20 min). The clarified lysate was incubated at 60 °C for 20 min and centrifuged again (4 °C; 10,000× *g*; 20 min) to remove denatured *E. coli* proteins. Purification of the His-tagged recombinant protein was conducted using a HiTrap^TM^ TALON column (GE Healthcare, Uppsala, Sweden). The protein was eluted with 10 mL of buffer B (50 mM NaH_2_PO_4_ (pH 8.0), 500 mM NaCl, 0.1% [*v*/*v*] Triton X-100, 10% [*v*/*v*] glycerol) containing 200 mM imidazole. Enzyme-containing fractions were pooled and dialyzed against buffer C (20 mM Tris-HCl (pH 8.0), 50 mM NaCl, and 5% [*v*/*v*] glycerol). The dialyzed Tt72 DNA polymerase was further purified by loading it onto a HiTrap^TM^ Heparin HP column (GE Healthcare, Uppsala, Sweden) pre-equilibrated with buffer C. The column was washed with the same buffer, and the bound protein was eluted using a linear gradient of 50–1000 mM NaCl in buffer D (20 mM Tris-HCl (pH 8.0), 1 M NaCl, and 5% [*v*/*v*] glycerol). Fractions containing Tt72 DNA polymerase were pooled and dialyzed against storage buffer (20 mM Tris-HCl (pH 8.0, 50 mM NaCl, 1 mM DTT, 0.1 mM EDTA, and 50% [*v*/*v*] glycerol) and stored at −20 °C. The purity of Tt72 DNA polymerase at each purification step was assessed by SDS-PAGE. The concentration of purified protein was determined using the Bradford assay [55].

### 4.3. 3′→5′ Exonuclease Assay

The 3′ end-labeled DNA substrate was prepared as previously described [30]. Reaction mixtures (50 µL) contained 10 mM Tris-HCl (pH 8.5), 25 mM KCl, 0.5 mM MgCl_2_, 10 mM (NH_4_)_2_SO_4_, 0.5 µg of the labeled substrate (λ/HindIII) and 0.8 μg of Tt72 DNA polymerase. The samples were incubated at 55 °C for 10 min, either in the presence or absence of dNTPs. The reactions were subsequently quenched by the addition of 200 µL of 10% [*w*/*v*] trichloroacetic acid (TCA). The reaction products were applied to GF/C filter disks (Whatman) and washed four times with 5 mL of 5% [*w*/*v*] TCA, followed by two washes with 5 mL of 70% [*v*/*v*] ethanol. The filters were air-dried and counted using a liquid scintillation counter (Perkin Elmer, Waltham, MA, USA) to determine the percentage of radioactivity released from the labeled DNA fragments. This assay was also employed to evaluate the effects of temperature (55–95 °C) and dNTP concentration (0–1000 µM) on the 3′→5′ exo activity.

### 4.4. DNA Polymerase Fidelity Assay

A plasmid-based *lacZ*α assay for DNA polymerase fidelity was performed as described in the original protocol [31]. The DNA polymerase fidelity reactions were conducted with Tt72 DNA polymerase, while Pfu and Taq DNA polymerases were included as control enzymes. Plasmid gap-filling reactions were prepared in a 20 µL mixture containing 10 mM Tris-HCl (pH 8.5), 25 mM KCl, 0.5 mM MgCl_2_, 10 mM (NH_4_)_2_SO_4_, 250 µM of each of four dNTPs, 1 nM gapped pSJ3 plasmid and 100 nM of enzyme. After incubation for 30 min at 55 °C, the extension products were introduced into *E. coli* DH5*α*. Transformants were selected on LA plates supplemented with 100 µL ampicillin, 20 µg/mL X-Gal, and 100 µM IPTG and subsequently scored for blue/white colonies. Background mutations were assessed by either omitting DNA polymerase from the reaction or directly introducing gapped pSJ3 into *E. coli* cells. A background mutation rate of 4.68 × 10^−4^ was used for gapped pSJ3. Mutant frequency and error rate were calculated as previously described [31,32]. Finally, white colonies (indicating *lacZ*α mutants) were subjected to DNA sequencing (Genomed, Warsaw, Poland) for further analysis.

### 4.5. Preparation of pUC18 Linear Form

Plasmid pUC18 was digested with either PstI (to generate 3′-overhangs), BamHI (to produce 5′-overhangs), or SmaI (to generate blunt ends). The reactions were conducted at 37 °C for 1 h in the buffer recommended by the supplier (Thermo Fischer Scientific). Following digestion, the linearized plasmid was purified using gel electrophoresis on a 1% agarose gel, stained with ethidium bromide, and visualized under UV light. The band corresponding to the linearized plasmid was extracted with the Gel-Out AX kit (A&A Biotechnology, Gdansk, Poland). The purified DNA was then stored at –20 °C for future use.

### 4.6. Filling in the 5′-Overhangs

The filling-in reactions were performed with Tt72 DNA polymerase. Control experiments were conducted with three additional enzymes: Klenow Fragment, Pfu, and T4 DNA polymerase using buffers recommended by manufacturers. The Tt72 DNA polymerase reaction mixtures (50 μL) contained Tt72 buffer (10 mM Tris-HCl (pH 8.5), 50 mM KCl, 0.5 mM MgCl_2_, 20 mM (NH_4_)_2_SO_4_) supplemented with 100 µM of each dNTP, 100 ng pUC18/BamHI and 0.8 μg of the Tt72 DNA polymerase. Reactions were incubated at 55 °C for 10 min. Then, the processed plasmid DNA was purified using a Clean-Up AX kit (A&A Biotechnology, Gdansk, Poland) and, after ligation, used to transform *E. coli* DH5α cells. Transformants were selected on LA plates containing 20 µg/mL X-gal, 20 µg/mL IPTG, and 100 µg/mL ampicillin. The plates were incubated at 37 °C for 18 h, and colonies were scored based on blue/white screening.

### 4.7. Removal of the 3′ Overhangs

The reaction to remove 3′-overhangs was conducted with Tt72 DNA polymerase. Control experiments were performed with two additional enzymes: Klenow Fragment and T4 DNA polymerase using buffers recommended by manufacturers. For Tt72 DNA polymerase reaction mixtures (50 μL) contained Tt72 buffer (10 mM Tris-HCl (pH 8.5), 50 mM KCl, 0.5 mM MgCl_2_, 20 mM (NH_4_)_2_SO_4_) supplemented with 100 µM of each dNTP, 100 ng pUC18/PstI and 0.8 μg of the Tt72 DNA polymerase. Reactions were incubated for 10 min at 55 °C. Then, the processed plasmid DNA was purified with a Clean-Up AX kit and, after ligation, used to transform *E. coli* DH5α cells. Transformants were selected on LA plates containing 20 µg/mL X-gal, 20 µg/mL IPTG, and 100 µg/mL ampicillin. The plates were incubated at 37 °C for 18 h, and colonies were scored based on blue/white screening.

### 4.8. 3′-Ends Polishing of PCR Products

The cassette carrying the *Cam^R^* gene was PCR-amplified using Taq DNA polymerase with pHSG576 as a template. Due to the terminal transferase activity of the enzyme, the PCR product possessed an additional nucleotide at the 3′ ends. The amplification was conducted using the following primers: forward: 5′-ACTGCTTCCGGTAGTCAATAAAC-3′ and reverse: 5′-ACAACTTTTGGCGAAAATGAGAC-3′. The reaction mixture included 1x Taq buffer (A&A Biotechnology), 1 ng of plasmid DNA, 0.1 µM of each primer, 250 µM of each dNTP, and 1U of Taq DNA polymerase. The cycling protocol consisted of an initial denaturation at 95 °C for 5 min, followed by 30 cycles of 30 s at 95 °C, 30 s at 53 °C, and 45 s at 72 °C. A final extension step was performed at 72 °C for 5 min. The amplified *Cam^R^* cassette was purified using a Gel-Out AX kit (A&A Biotechnology, Gdansk, Poland).

The polishing of the 3′-ends of the PCR product was carried out using Tt72 DNA polymerase purified in our laboratory. For control experiments, three other enzymes were tested: Klenow Fragment, Pfu, and T4 DNA polymerase. The reaction mixtures contained Tt72 buffer (10 mM Tris-HCl (pH 8.5), 50 mM KCl, 0.5 mM MgCl_2_, 20 mM (NH_4_)_2_SO_4_) supplemented with 100 µM dNTP, 500 ng PCR-amplified *Cam^R^* cassette, and 0.8 μg of the Tt72 DNA polymerase. The reactions were incubated at 55 °C for 10 min. The processed *Cam^R^* cassette was then purified with a Clean-Up AX kit and cloned into pUC18 linearized with SmaI. The ligation mixture was used to transform *E. coli* DH5α cells. Transformants were plated on LA plates containing 20 µg/mL X-gal, 20 µg/mL IPTG, and 100 µg/mL ampicillin. The plates were incubated at 37 °C for 18 h and scored for blue/white colonies.

## Figures and Tables

**Figure 1 ijms-25-13544-f001:**
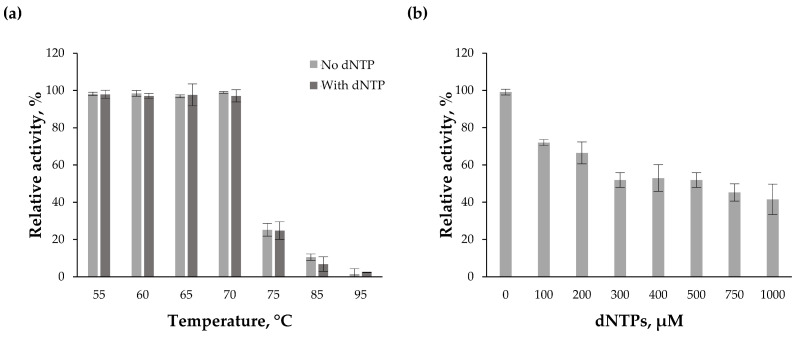
Characterization of 3′→5′ exonuclease activity of the Tt72 DNA polymerase. The enzyme’s activity was assessed by measuring the radioactivity released from labeled λ/HindIII DNA fragments. (**a**) 3′→5′ exo activity of Tt72 DNA polymerase after 10 min heat treatment at temperatures ranging from 55 °C to 95 °C. (**b**) Effect of dNTP concentration on the 3′→5′ exo activity of the Tt72 DNA polymerase. Error bars indicate the mean  ±  standard deviations. Each experiment was conducted in triplicate.

**Figure 2 ijms-25-13544-f002:**
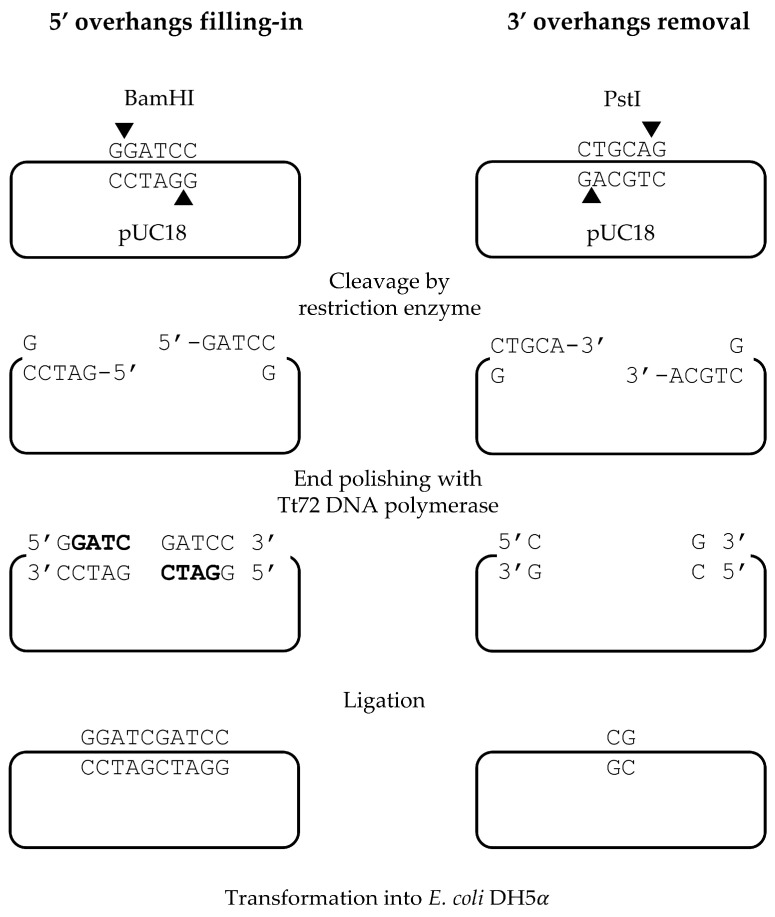
Schematic illustration of DNA 3′/5′-ends polishing with Tt72 DNA polymerase. Cutting sites are indicated by triangles.

**Figure 3 ijms-25-13544-f003:**
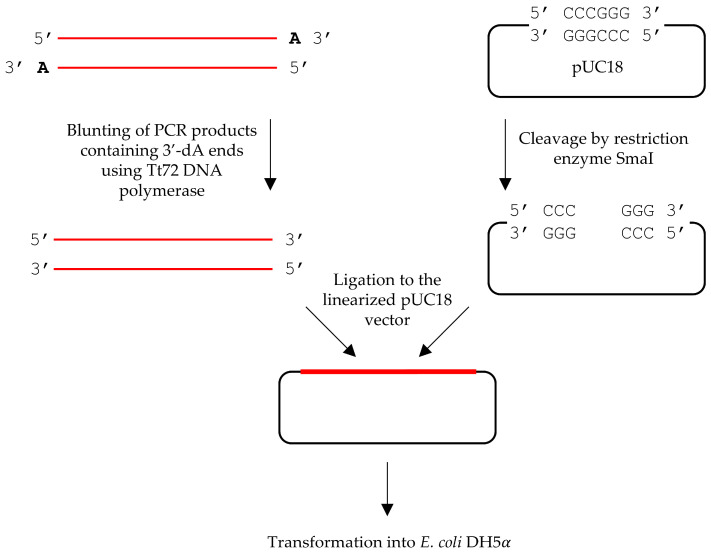
Schematic illustration of PCR product 3-overhang polishing with Tt72 DNA polymerase.

**Table 1 ijms-25-13544-t001:** Fidelities observed for DNA polymerases assessed through a plasmid-based *lacZ*α gene complementation assay.

Polymerase	Total Number of Colonies ^1^	Number of White Colonies	Mutation Frequency ^2^	Error Rate ^3^
Tt72 DNA polymerase	26,884	68	2.06 × 10^−3^	1.41 × 10^−5^
Tt72 DNA polymerase exo^−^ (D78A)	17,682	119	6.23 × 10^−3^	4.29 × 10^−5^
Taq DNA polymerase	39,661	266	6.24 × 10^−3^	4.27 × 10^−5^
Pfu DNA polymerase	28,866	30	5.72 × 10^−4^	3.91 × 10^−6^

^1^ The colony counts represent the sum of three independent experiments, each performed with four replicates. ^2^ The mutation frequency was calculated using the formula: ([number of white colonies/total number of colonies] − background mutation rate). A background mutation rates of 4.68 × 10^−4^ were used for gapped pSJ3. ^3^ The error rate, defined as the number of errors per base incorporated, was determined from the mutation frequency using previously established methods [31].

**Table 2 ijms-25-13544-t002:** Polymerase polishing of 5′-overhang of pUC18/BamHI.

Polymerase	Number of Colonies ^1^	Number of White Colonies ^1^	Efficiency (%) ^2^
Tt72 DNA polymerase	14,889	12,749	85.7 ± 0.90
T4 DNA polymerase	14,055	12,774	90.9 ± 0.23
Klenow Fragment	10,404	8237	79.8 ± 1.57
Pfu DNA polymerase	13,947	10,649	76.6 ± 1.69

^1^ The number of blue and white colonies is summed from three independent experiments. ^2^ The ratio of white to total colonies reflects the efficiency of the dsDNA end-blunting process.

**Table 3 ijms-25-13544-t003:** Polymerase polishing 3′-overhang of pUC18/PstI.

Polymerase	Number of Colonies ^1^	Number of White Colonies ^1^	Efficiency (%) ^2^
Tt72 DNA polymerase	15,468	14,944	96.6 ± 0.17
T4 DNA polymerase	12,606	12,294	97.5 ± 0.04
Klenow Fragment	10,132	7800	77.0 ± 1.35

^1^ The number of colonies and white colonies is summed from three independent experiments. ^2^ The ratio of white to total colonies reflects the efficiency of the dsDNA end-blunting process.

**Table 4 ijms-25-13544-t004:** Polymerase polishing of PCR DNA fragments.

Polymerase	Number of Colonies ^1^	Number of White Colonies ^1^	Efficiency (%) ^2^
Tt72 DNA polymerase	19,589	1458	7.4 ± 0.62
T4 DNA polymerase	10,525	797	7.6 ± 0.25
Klenow Fragment	23,912	631	2.6 ± 0.40
Pfu DNA polymerase	20,876	1100	5.3 ± 0.59

^1^ The total number of colonies and white colonies represents the combined results from three independent experiments. ^2^ The ratio of white to total colonies serves as an indicator of the efficiency of PCR product end polishing.

## Data Availability

The raw data supporting the conclusions of this article will be made available by the authors on request.

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
