# Peer review of "Precise and Accurate DNA-3′/5-Ends Polishing with Thermus thermophilus Phage vb_Tt72 DNA Polymerase"

_ijms, 2024, doi:10.3390/ijms252413544_

Round 1

Reviewer 1 Report

Comments and Suggestions for Authors In this manuscript, the authors do a nice job characterizing a previously undescribed Thermus thermophilus phage DNA polymerase (vb_Tt72 DNA polymerase). Given the lack of information and the relevance for such polymerases in molecular biology applications, this work is potentially important and of interest to a variety of researchers. This work adds to an existing body of research on related DNA polymerases and could be useful for researchers interested in improving molecular cloning methodologies, which has broad applications in the biomedical sciences. The conclusions of the manuscript are well-supported by the presented data, the authors were successful in addressing the main question that they posed on the function of this novel DNA polymerase and in describing how this DNA polymerase could be useful in molecular cloning applications. The cited references are appropriate for this manuscript. I do not have any concerns with the manuscript in terms of the background, the experimental methods that were used, or in the conclusions that the authors draw from their findings. However, on Pg 1, lines 30-31, it may be more accurate to say the "repair of double strand breaks via NHEJ and HR" than "repair of recombination-induced damage...". NHEJ repairs double-strand breaks, not necessarily recombination intermediates.

Author Response

Reviewer 1

We appreciate the reviewer's feedback and positive assessment of our manuscript.

Comments 1: "However, on Pg 1, lines 30-31, it may be more accurate to say the "repair of double strand breaks via NHEJ and HR" than "repair of recombination-induced damage...". NHEJ repairs double-strand breaks, not necessarily recombination intermediates."

Response 1: We have followed the reviewer's suggestion and revised lines 30-31, particularly the sentence concerning joining non-homologous ends.

Reviewer 2 Report

Comments and Suggestions for Authors

Dorawa and Kaczorowski present the characterisation of the 3'->5' exonuclease activity of their previously reported Tt72 DNA polymerase. The report uses well-established methodologies that have been phased out by the field in recent years. Nonetheless, the experiments are well performed and add information on Tt72 to the literature.

I have a few comments which I put in roughly the order they appear. More significant comments are highlighted (*).

1. Fraction of active polymerase. Unfortunately, the authors did not quantify what fraction of the polymerase was active. 

2.* While end-polishing does give a measure of the exonuclease activity of the enzyme, it is biased towards extremes, i.e. no polishing or complete polishing. Incomplete polishing tends to generate constructs that do not ligate efficiently and therefore are unlikely to transform. As a result, end-polishing experiments overestimate the end-polishing activity of Tt72. Maybe better would be to carry out a primer extension assay where exonuclease could be assessed quantitatively.

3.* Mutation bias. The sampling of 15 or 20 clones from white colonies (i.e. including only mutations that are non-synonymous and that cannot be tolerated) is very low to start a discussion on the mutation spectra of the enzyme. Better would have been to have used NGS where even a small experiment of 50k short products would give a much better insight into the error rate of the enzyme.

4. Background mutation rates described in the M&M (L.321) are not given in the article. They must be included.

5.* Amount of enzyme in end-blunting reactions. By my calculations, the molar amounts of enzyme (~10pmol) used in the fill-in and end-polishing exceeds that of the template (~57fmol). I don't think it is possible to talk about efficiency under these conditions.

6. L240. I think a 20-year-old paper is no longer recent (ref.50). 

7. What would be an interesting question is what fraction of the PCR amplicon is devoid of the 3'-A/G that PolAs often incorporate. Does Tt72 deliver fewer overhangs than Taq? Is that comparable with B-family polymerases that leave only trace amounts of 3'-purine overhangs?

8. Purity of the polymerase was assessed but never given in the manuscript.

Author Response

Our response to Reviewer 2's comments is attached as a PDF file.
